Smart waste management and classification system using advanced IoT and AI technologies

http://orcid.org/0000-0001-6794-3677 Alourani Abdullah 1
Ashraf M. Usman 2 usman.ashraf@gcwus.edu.pk
Aloraini Mohammed 3
1 Department of Management Information Systems, College of Business and Economics, Qassim University , Buraydah , Saudi Arabia
2 Computer Science, GC Women University Sialkot , Sialkot, Punjab , Pakistan
3 Department of Electrical Engineering, College of Engineering, Qassim University , Buraydah , Saudi Arabia
Angiulli Giovanni
Electronic publication date: 2025 Apr 1
Publication date: 2025
Volume: 11
Electronic Location ID: e2777
Received 2024 Oct 28; Accepted 2025 Feb 28
Copyright: © 2025 Alourani et al.
Copyright year: 2025
Copyright holder: Alourani et al.
License: This is an open access article distributed under the terms of the Creative Commons Attribution License, which permits unrestricted use, distribution, reproduction and adaptation in any medium and for any purpose provided that it is properly attributed. For attribution, the original author(s), title, publication source (PeerJ Computer Science) and either DOI or URL of the article must be cited.
License URL: https://creativecommons.org/licenses/by/4.0/

Keywords: Waste management system, Industrial Internet of Things, Solid waste recycling, Cloud manufacturing service, Intelligent manufacturing system, Artificial intelligence, Machine learning

Funding: Qassim University, Deanship of Scientific Research 2023-SDG-1-HSRC-36995 This work was supported by Qassim University, represented by the Deanship of Scientific Research, on the financial support for this research under the number (2023-SDG-1-HSRC-36995) during the academic year 1445 AH / 2023 AD. The funders had no role in study design, data collection and analysis, decision to publish, or preparation of the manuscript.

==============================
The effective management of municipal solid waste is a critical global issue, affecting both urban and rural areas. To address the growing volume of solid waste, proactive planning is essential. Traditionally, solid waste is often disposed of without segregation, preventing recycling and the recovery of raw materials. Proper waste segregation is a fundamental requirement for effective solid waste management, allowing materials to be recycled efficiently. Emerging technologies such as artificial intelligence (AI), machine learning (ML), and the Internet of Things (IoT) offer powerful tools for identifying recyclable materials like glass, plastic, and metal within solid waste. The primary goal of this research is to contribute to a cleaner environment, reduce infant mortality, improve maternal health, and support efforts to combat HIV/AIDS, malaria, and other diseases. This study introduces an intelligent and smart solid waste management system (iSSWMs) designed to smartly collect and segregate solid waste. The proposed system focuses on three types of materials: plastic, glass, and metal. The first phase involves waste collection using smart bins connected to a mobile application, which sends notifications when the bins are full. In the second phase, we develop a deep learning-based mechanical model to segregate the waste, using the VGG-19 model, which achieved a performance accuracy of 99.7% during training. To the best of our knowledge, iSSWMs is a promising framework that integrates both waste collection and segregation through the use of cutting-edge technologies, delivering high accuracy and efficiency.

Introduction

Solid waste management encompasses the systematic collection, transportation, processing, recycling, and disposal of solid waste generated by human activities. It plays a crucial role in ensuring environmental sustainability and safeguarding public health (Nanda & Berruti, 2021; Nemerow et al., 2009). The process typically starts with waste separation at its source, followed by transportation to treatment facilities or landfills (Vinti et al., 2021; Ziraba, Haregu & Mberu, 2016). Key components of effective waste management include recycling and composting, which reduce the volume of waste sent to landfills and help minimize environmental impact (Jayakrishnan, Jeeja & Bhaskar, 2013). Additionally, waste-to-energy initiatives have become increasingly important, offering a means to extract energy from certain types of solid waste.

Sustainable and comprehensive solid waste management practices are vital for reducing pollution, conserving resources, and fostering cleaner, healthier communities (Abubakar et al., 2022). Achieving this requires collaboration among governments, communities, and individuals to promote responsible waste disposal and encourage recycling (Sharma et al., 2021). However, several challenges hinder effective solid waste management, including environmental, social, and economic factors. A major challenge is the collection and segregation of recyclable waste, especially in rapidly urbanizing areas (Babazadeh et al., 2020; Joshi & Ahmed, 2016). Insufficient infrastructure for waste collection, transportation, and disposal further exacerbates the issue, leading to inefficient waste management systems. Improper disposal methods, such as illegal dumping and open burning, contribute to environmental degradation and health risks (Goswami, 2018).

Recycling and waste separation efforts are often limited by a lack of awareness and inadequate facilities, while financial constraints, particularly in developing regions, hinder investments in advanced waste management technologies. The growing global population and urbanization place additional strain on existing waste systems (Ahsan et al., 2015), underscoring the need for innovative, sustainable solutions (Farooq et al., 2022; Nižetić et al., 2019; Shekdar, 2009). Balancing economic factors, environmental impact, and social equity in waste management remains a persistent challenge, one that requires coordinated efforts, policy interventions, public awareness campaigns, and investments in infrastructure to create more efficient and environmentally friendly systems (Noor et al., 2020; Sodiq et al., 2019).

In this article, we propose a novel artificial intelligence (AI) and Internet of Things (IoT)-based intelligent and smart solid waste management system (iSSWMs) to improve waste collection and segregation. In the first phase, we developed a smart dustbin connected to IoT devices, which transmits specific data (such as bin capacity and location) to a cloud server once it reaches capacity (Han et al., 2012). The cloud server then sends an automatic alert to waste collectors with the bin’s information, enabling efficient waste collection. In the second phase, we employ machine learning (Atayero et al., 2019; Catarinucci et al., 2020) to train the system for waste segregation. The system randomly selects an item from a heap of waste, moves it along a conveyor belt, accurately recognizes it using AI-based image processing, and places it into the appropriate bin. The segregated raw materials are then delivered to waste management companies for recycling into new products.

The primary objective of the proposed iSSWMs model is to contribute to reducing infant mortality, improving maternal health, and combating HIV/AIDS, malaria, and other diseases. In developing countries, the growing volume of waste and inadequate waste management practices are significant contributors to the spread of communicable diseases. Our approach focuses on waste collection, utilizing advanced IoT technologies and mobile applications, as well as segregation using convolutional neural network (CNN)-based models for image recognition and an actuator mechanism on the conveyor belt. A camera mounted on the conveyor belt identifies waste items, relaying the information to a connected Raspberry Pi (RP), which then signals actuators to sort the waste into designated bins.

This methodology encompasses the initial classification of waste into four categories including plastic, glass, metal, and trash (mixed waste) categories. Overall, the system is implemented in a controlled environment featuring a designated dumping unit. Further our contribution can be summed up as follows: Devise an intelligent and smart solid waste management system (iSSWMs) that collect and segregate the solid waste smartly by using the advanced emerging technologies such as IoT, cloud computing, deep learning, android mobile application, and actuator system.

Development of smart bins to be used for solid waste (plastic, metal, glass) collection smartly. Which are further linked with ThingSpeak server for data analytics and visualization in the cloud.

Development of Android mobile application to get the bins information, and alert to collect the waste from specific location once the bin is filled according to predefined threshold values.

Use TashNet dataset and train the model using CNN VGG-19 for waste image recognition.

Deploy multiple actuators on conveyor belt controlled by micro-controller and object detecting sensor(s).

Rest of the article is organized in such way that “Literature Review” presents a comprehensive literature study explain existing state-of-the-art methods on solid waste management. In “iSSWMs Model”, we present our proposed iSSWMs framework in details. Further “Implementation and Results” describes the implementation of proposed iSSWMs model, and results comparing with existing state of the art methods. Finally, “Conclusion” conclude the study.

Literature review

We now give an overview of background knowledge of used emerging technologies. Further we present the state-of-the-art techniques used in waste management and segregation in “Related Work”.

Technology background

Mobile computing

Mobile applications play a crucial role in providing alerts in suspicious cases, contributing to enhanced efficiency and convenience in various aspects of daily life (Elazhary, 2019; Stergiou et al., 2018). One notable example is the integration of smart waste management systems with mobile applications. In this context, sensors installed in waste bins can detect the fill level and relay this information to a connected mobile app. For instance, consider a scenario where a smart waste bin is equipped with sensors to monitor its capacity. When the bin reaches a predefined threshold, indicating that it is nearly full, the mobile application linked to the waste management system generates an automatic alert. This alert is then sent to relevant authorities, waste collection services, or even individual users, notifying them that the bin requires attention and timely emptying. Such applications not only streamline waste collection processes but also contribute to more efficient resource management. Users, whether municipal waste management teams or individual households, can receive timely alerts, preventing overflows and ensuring that waste collection services are deployed proactively when needed. This integration of mobile applications with IoT-enabled devices exemplifies how technology can optimize routine tasks, improve environmental sustainability, and enhance overall urban living (Pardini et al., 2020; Wang et al., 2021).

Internet of Things

The IoT has become an integral part of daily life, seamlessly intertwining with various aspects to enhance convenience, efficiency, and connectivity. In our homes, smart devices like thermostats, lights, and security cameras are interconnected through IoT, allowing us to control and monitor them remotely using smartphones or voice commands. Wearable devices, another facet of IoT, help individuals track their health metrics, exercise routines, and sleep patterns, providing valuable insights for personal well-being. In urban settings, IoT contributes to smart cities by managing traffic flow, optimizing energy consumption, and enhancing public services (Shyam, Manvi & Bharti, 2017). In agriculture, IoT sensors monitor soil conditions, crop health, and weather patterns, aiding farmers in making informed decisions (Singh et al., 2022). Retail experiences are also transformed through IoT with inventory management, personalized shopping recommendations, and contactless payments. The omnipresence of IoT in daily life not only increases efficiency but also fosters a more connected and responsive environment, heralding a future where the seamless integration of technology continues to redefine our day-to-day experiences (Ali et al., 2020).

Similarly, IoT play a vital role in solid waste management in term of transformative, revolutionizing traditional practices and enhancing the overall efficiency of waste-related processes. IoT technologies introduce a network of interconnected sensors, devices, and data analytics tools into the waste management ecosystem (Patil & Kale, 2016). These innovations facilitate real-time monitoring, data-driven decision-making, and improved resource allocation. Smart waste bins equipped with IoT sensors can relay information on their fill levels, enabling optimized waste collection routes and schedules. This not only minimizes operational costs but also contributes to reduced fuel consumption and environmental impact. Predictive maintenance capabilities ensure the health of waste collection vehicles, minimizing downtime and optimizing fleet management. Environmental monitoring through IoT devices provides valuable insights into the impact of waste management activities on the surroundings, aiding in sustainable and ecologically responsible practices. Additionally, IoT fosters greater citizen engagement by offering real-time information on waste disposal locations, schedules, and recycling guidelines through user-friendly applications (Dubey, Patel & Kumar, 2020). In essence, the integration of IoT in waste management systems leads to a smarter, more responsive, and environmentally conscious approach to handling and mitigating the challenges of waste in our communities.

Deep learning

Under the umbrella of artificial intelligence, deep learning plays a crucial role in waste management systems by significantly enhancing waste object detection, a key aspect of efficient waste sorting and recycling processes. With the increasing complexity of waste composition, accurate identification and classification of waste objects are vital for optimizing recycling workflows. Deep learning techniques, CNNs, have demonstrated remarkable capabilities in image recognition and object detection (Nait Aicha, Archambault & Bouchard, 2018; Schuller, 2015). In waste management, deep learning models are trained on extensive datasets containing images of various waste items. These models learn to recognize specific objects and materials, enabling them to accurately identify and classify items such as plastics, metals, article, and organic waste. Integrated with sensors and cameras in waste sorting facilities, these deep learning-based systems can analyze images in real-time, facilitating automated sorting processes with unprecedented accuracy. The advantages of deep learning for waste object detection extend beyond traditional recycling facilities. Smart waste bins equipped with cameras and sensors utilize deep learning algorithms to identify and categorize items as they are disposed of. This real-time analysis helps improve waste collection efficiency, optimize recycling efforts, and reduce contamination in recycling streams (Hasan, Alam & Jang, 2022; Sheng et al., 2020; Kumar, Yadav & Singh, 2021). By leveraging deep learning for waste object detection, waste management systems can achieve higher precision in sorting, reduce reliance on manual labor, and enhance overall sustainability by promoting more effective recycling practices. As technology continues to advance, the integration of deep learning in waste management promises continuous improvements in waste object detection accuracy and contributes to the development of smarter and more environmentally friendly waste management solutions (Rahman, Debnath & Khan, 2022; Wang et al., 2021; Jadli & Hain, 2020).

Related work

In an age characterized by the rapid modernization of all societal sectors, technology has become an integral aspect of daily life for individuals. The profound alterations in human lifestyles, propelled by technological advancements, are observable as modern technology permeates every facet of human existence. Cities endowed with cutting-edge facilities and technology are preferred, leading to a continual surge in urban population, accompanied by both advantages and disadvantages. The increasing congestion in growing urban areas brings about worries concerning health, safety, and environmental issues, which encompass aspects such as medical services, security, privacy, and transportation (Javed et al., 2022).

Solid waste management emerges as a notable challenge in contemporary urban areas, involving intricate processes encompassing to collection and transferring at particular dump area, classification, and recycling the segregated solid waste. The heightened utilization of natural resources has resulted in alarming levels of waste, posing threats to human health and the environment. Despite the awareness among the educated population in urban areas regarding the environmental impact of waste, a significant portion disposes of waste without proper sorting and recycling.

Recycling assumes a central role as evidenced by studies indicating that nearly 0.75% of solid domestic waste can be recycled, presenting economic benefits and environmental sustainability. Countries worldwide, including America, China, Japan, Canada, Korea, Russia, Italy, Malaysia, Saudi Arabia, and Qatar, actively engage in developing smart waste management systems (Fidje, Haddara & Langseth, 2023). Previous endeavors involve techniques implemented in cities like St. Petersburg, Russia, utilizing technologies such as wireless sensor networks (WSNs), radio frequency identification (RFID), sensors, and actuators (Ogarkov, 2019; Vishnu et al., 2021).

Adedeji & Wang (2019) introduced a novel waste management system tailored for smart cities, harnessing the capabilities of the IoT to overcome limitations inherent in conventional waste management systems. The proposed framework consists of two distinct types of end sensor nodes: public bin level monitoring units (PBLMUs) designed for monitoring public bins, and home bin level monitoring units (HBLMUs) intended for residential areas. Subsequently, they process this information and transmit it to a centralized monitoring station for storage and analysis. The experiments were conducted to assess the power consumption of a PBLMU and estimate its life expectancy under hypothetical conditions, providing valuable insights into the system’s practicality and efficiency. The data extracted from the IoT system, including trash bin details such as unfilled level and geolocation coordinates, can be employed to create a geographic information system (GIS). Additionally, ML algorithms can be applied to determine optimal routes for waste collection trucks as well as segregation of the waste.

In the domain of waste classification, various image classification techniques, including support vector machine (SVM), CNN, and ResNet-50, have been employed (Lilhore et al., 2024). The rise of deep learning models, particularly CNN architectures like VGG16 (Ahmed et al., 2023) and VGG19 (Vaidya & Paunwala, 2019; Bansal et al., 2023), has demonstrated increased effectiveness for object detection and classification compared to traditional methods. The incorporation of IoT technologies into waste management systems has given rise to the development of intelligent and automated waste management solutions in smart cities, thereby enhancing efficiency and sustainability.

Leading to IoT and emerging technologies, Shahab & Anjum (2022) worked on sustainable waste management and proposed a novel AI and IoT based multi-path (MP) convolutional neural network model. The proposed study was particularly for delhi region in India. They defined the category into two major classes including waste and non-waste. Further waste category is dumped at specific region and implement waste classification. According to the authors, the proposed MP-CNN model achieved 98% accuracy in waste detection while segregating process, however it was tested in short volume data with just 600 images. The same mechanism was proposed by Krishna & Sharma (2023) with improved datasets containing 1,150 images. They used multiple classifiers including VGG-16, ResNet-16 etc., but their performance was not up to the mark and achieved 93% accuracy. To improve the waste collection process, Sheng et al. (2020) implemented LoRa protocols for better communication while collecting waste from smart dustbins. According to authors the communication channels should be stronger to provide the correct information and alerts when a bin is filled and supposed to be dumped for further segregation. Indeed the proposed study achieved remarkable outcome and improved the waste collected process using advanced IoT technologies, however, waste segregation was not addressed in their proposed study. Below is a comparative analysis of existing state of the art proposed studies presented in Table 1.

Table 1 A comparative analysis of existing studies proposed for waste collection and segregation using IoT and AI technologies.

Ref. no	Model name	Significance of model	Used dataset	Accuracy	Limitations	
Adedeji & Wang (2019)	IoT-based waste management system (PBLMU & HBLMU)	Uses IoT to monitor and manage waste bins, integrates GIS and ML for route optimization	Simulated IoT data	Not specified	Focuses on waste collection but lacks automated segregation	
Lilhore et al. (2024)	SVM, CNN, ResNet-50	Image classification techniques for waste categorization	Not specified	Varies (higher accuracy with CNN models)	Limited dataset, requires real-world validation	
Ahmed et al. (2023)	VGG16	Deep learning-based waste classification	Not specified	Higher accuracy than traditional methods	Computationally expensive, needs large datasets	
Vaidya & Paunwala (2019)	VGG19	Improved deep learning model for waste detection	Not specified	Higher accuracy than VGG16	Requires more computational power	
Shahab & Anjum (2022)	MP-CNN (Multi-path CNN)	AI & IoT-based model for waste classification (Delhi region)	600 images	98%	Limited dataset size, requires scalability	
Krishna & Sharma (2023)	Multiple classifiers (VGG-16, ResNet-16)	Waste classification using improved dataset	1,150 images	93%	Performance not optimal, further enhancements needed	
Sheng et al. (2020)	IoT with LoRa protocols	Enhances waste collection efficiency using smart dustbins	Not specified	Not specified	Lacks waste segregation mechanisms	

Though several techniques have been proposed over sustainable waste collection and segregation process as discussed above, however, it has been observed that the implementation ibn real environment is still a challenge being face worldwide. To cope this challenge, this study proposed a novel scheme that deal both waste collection mechanism as well as segregation discussed in following section.

iSSWMs model

The proposed iSSWMs comprised of multiple elements including smart bins, cloud processing and storage, mobile application, waste collector image capturing and recognition, and actuator pump. The working of iSSWMs divided into two phases. The detail workflow of proposed iSSWM is presented in “Performance Evaluation Metrics” followed by brief overview of iSSWM elements as follows.

iSSWMs elements

Smart bins

The smart dustbin, equipped with advanced sensors such as weight and volume sensors, revolutionizes waste collection by efficiently monitoring and managing waste levels. In this study, we have connected four bins including glass bin, plastic bin, metal bin, and trash bin (used for waste other than selected waste types). The controller receives the sensory information continuously from all the connected bins, and forward to cloud server along with bin location information for further processing and storage. Figure 1 presents the working principle of smart bin as follows:

Figure 1 Working principle of smart bin to collect waste items smartly.

Cloud processing and storage

In proposed iSSWMs, we used ThingSpeak that serves as a pivotal platform for orchestrating data flow and decision-making processes. It receives real-time data streams from IoT devices, particularly sensory data from smart bin controllers, which monitor parameters such as waste volume, weight, and other relevant metrics. Upon receiving this data, ThingSpeak’s analysis capabilities come into play, processing the incoming sensory information to detect patterns, trends, and predefined thresholds indicative of critical events, such as when waste accumulation reaches predetermined levels. At the same time, the analyzed data is forwarded on real time cloud storage “Firebase database” which is further connected with android mobile application to receive the analytics on smart devices. Once these thresholds are crossed, ThingSpeak initiates actions to notify over mobile application through broadcasting alert message(s) that is ultimately received to relevant stakeholders, signaling the need for intervention.

Waste collector

Upon receiving the waste information notification, the waste collector promptly proceeds to the designated bin location to collect the accumulated waste. Upon arrival, the waste collector unloads the collected waste from the smart bins and transports it to a predetermined dump area designated for further segregation processes. This meticulous process ensures that the collected waste is efficiently managed and prepared for subsequent stages of waste treatment and disposal.

Image capturing and recognition

At this stage, the waste segregation process commences with the waste being positioned on a conveyor belt, ensuring that only one item moves forward at a time for recognition. At the starting edge of the conveyor belt, a camera configured to capture the image of the waste item in its immediate vicinity employs the highly accurate deep learning VGG-19 classification model. VGG-19 is renowned for its exceptional performance in image classification tasks. Developed by the Visual Geometry Group at the University of Oxford, VGG-19 features a deep CNN composed of 19 layers, comprising 16 convolutional layers and three fully connected layers. VGG-19 adeptly extracts pertinent features pivotal for precise classification. It stands out for its simplicity and effectiveness, employing small (3 × 3) convolutional filters and max-pooling layers to extract hierarchical features from input images. VGG-19 has demonstrated remarkable accuracy on benchmark image classification datasets, making it a popular choice for various computer vision applications. Renowned for its precision in waste categorization, VGG-19 ensures reliable identification of the waste item type, whether glass, plastic, or metal. Upon successful identification of the waste item type, the waste information is then transmitted to an attached controller responsible for activating the actuator pump functionality accordingly.

Actuator pump

At the final stage of segregation, the controller triggers the appropriate actuator pump (AP) based on the received waste type. To confirm that waste has arrive in front of the AP, we use Ultrasound sensor that detect the wave sound of any objective coming toward this. On arrival of waste item at accurate position, signal broadcasted, and relevant AP push the waste item to its adjacent bin. Therefore, AP working decision is based on twofold information including (1) information from the controller about waste category (2) sensor signal on arrival of waste item in front of it. Once the required information is received, relevant AP start working, and push the waste item to the adjacent bin through a slide. Conversely, if the waste item is not recognized properly, it is directed to a trash bin containing mixed waste, which is then sent for further segregation. In this systematic manner, recyclable waste is collected, analyzed, notified, and segregated, ensuring an efficient waste management process.

iSSWMs workflow

The proposed iSSWMs framework works in two phases. First phase is regarding the waste collection and data analytics. Further phase two deals with waste segregation using deep learning. Figure 2 shows the block diagram of proposed iSSWMs architecture. The detail workflow of iSSWMs architecture is as follows:

Figure 2 Smart bins to collect solid waste including (1) plastic, (2) glass, and (3) metal.

Phase 1: Intelligent bin monitoring and cloud-based analytics

In the first phase of the smart waste management system, the integration of cutting-edge technologies is strategically employed to optimize efficiency and promote sustainability. The process begins with the deployment of four intelligent bins, each designated for specific waste types: glass, plastic, metal, and organic waste. These smart bins are not merely receptacles; they are equipped with advanced sensors that continuously monitor critical parameters such as bin size, waste quantity, and precise bin location. The intelligence of these bins comes to the forefront when one of them reaches its capacity. At this point, the bin autonomously transmits a comprehensive set of data to the cloud. This data transmission includes real-time information about the bin’s size, the quantity of waste it contains, and its geographic location. The cloud-based database acts as the nerve center, swiftly processing and analyzing the incoming data. Subsequently, the relevant information is relayed to a dedicated mobile application, providing instant accessibility to all concerned authorities involved in waste management. Upon reception of the notification through the mobile application, designated personnel are promptly alerted to the specific location where waste needs to be collected. The collected waste is then systematically transported to a centralized dump area for further processing. This phase exemplifies the seamless coordination between intelligent bins, cloud-based analytics, and mobile applications, fostering a responsive and efficient waste collection process. A detailed procedure of waste collection and segregation is presented in Algorithm 1.

Algorithm 1 Formal steps of proposed iSSWMs for solid waste collection and segregation.

Declarations	
    •    SBs configure three smart bins	
    •    Bcap: Capacity of smart bin.	
    •   Bloc: Location of the smart bin	
    •   Bvol: Volume of waste in the smart bin	
    •   Bweight: Weight of waste in the smart bin	
    •   BID: Unique identifier of the smart bin	
    •   Wtype: Category of waste in the smart bin	
    •   APi: Actuators pump i	
Output:	
    •   msg: Alert message for waste collector	
    •   Witem: Waste category information for controller	
Procedure:	
    •   Configure three smart bins for glass, metal, and plastic waste collection.	
    •   For (Bvol < Bcap )	
    •   {	
    •   Receive waste in bin (s).	
    •   Transfer sensory data from smart bins to ThingSpeak for data analysis.	
    •   Forward analysed data from ThingSpeak to cloud storage (Firebase database).	
    •   Mobile application collects live data from Firebase database.	
    •   Broadcast alert if a bin is full and needs to be emptied.	
    •   Waste collector receives alert message on smart device and proceeds to bin location.	
    •   Waste collector unloads waste at waste dump area.	
    •   Waste is loaded onto a conveyor belt.	
    •   Configure a camera at the start of the conveyor belt to capture waste images.	
    •   Pass waste image information to a Deep Learning algorithm for analysis of waste category.	
    •   Deep Learning model forwards waste category information to the controller.	
    •   Controller is linked with three actuators pump (AP).	
    •   APi pushes waste into the relevant bin through a slide once the waste category is identified.	
    •   If waste category is not detected, place waste into a trash bin for segregation process.	
    •   Automatic system collects and segregates desired waste accordingly.	

Phase 2: Waste segregation (CNN)

The second phase of the smart waste management system delves into the intricacies of waste segregation, aiming to streamline the process through advanced automation. The waste collected in the central dump area is now subjected to a systematic segregation procedure. The waste items are conveyed on a sophisticated belt system, enhancing the efficiency of the overall process. As each item moves along the conveyor belt, a fixed camera captures detailed images of the waste. The integration of deep learning techniques, particularly CNN models, is instrumental in image recognition. The CNN models analyze the captured images, swiftly identifying the type of waste based on predetermined categories, such as glass, plastic, metal, or organic waste. The analysis information is forwarded to microcontroller which is controlling the attached three actuator pump placed on conveyor belt. Once the waste item start moving on conveyor belt, the relevant actuator pump push the waste item to perspective bin through connected slide. In this scenario, if the waste item is not recognized appropriately, the waste item is dropped in trash bin which is basically an irrelevant waste from the defined categories. This systematic and smart process segregate the waste items accurately with less power consumption and almost zero manpower. This automation significantly contributes to the promotion of recycling and environmentally responsible waste management practices. Figure 2 shows the block diagram of proposed iSSWMs.

For this study, the VGG-19 model was chosen due to its well-established performance in image classification tasks. VGG-19, originally trained on the large-scale ImageNet dataset, consists of deep convolutional layers capable of extracting hierarchical features from input images. Instead of training a deep learning model from scratch, which would require extensive computational resources and a larger dataset, transfer learning was employed by utilizing the pre-trained weights of VGG-19 and adapting them to the waste classification task.

The integration process of the proposed iSSWMs model ensures seamless connectivity between smart bins, cloud-based analytics, and automated waste segregation, making it highly scalable across diverse environments. The system’s modular design allows for easy deployment in urban, industrial, and commercial settings by adapting the number of smart bins and sensor configurations based on waste generation levels. The ThingSpeak platform enables real-time data processing and decision-making, facilitating effortless integration with existing waste management infrastructures. The cloud-based storage (Firebase) ensures scalability by efficiently handling data from multiple locations and providing remote access to stakeholders via mobile applications. Furthermore, the use of deep learning-based waste recognition, specifically VGG-19, enhances adaptability in different waste categorization scenarios, ensuring robustness even with varying waste compositions. The automated segregation process, driven by microcontroller-controlled actuator pumps, minimizes manual intervention and can be scaled by adding more conveyor lanes or actuator units for high-throughput environments. These design choices collectively support the system’s integration potential and scalability, ensuring efficient waste management across diverse operational landscapes.

Implementation and results

This section presents the implementation details and evaluates results. We implemented the proposed iSSWMs framework and existing studies in Python. The system platform and configurations used for the experiments are presented in Table 2.

Table 2 Implementations specifications.

Specifications	Details	
IDE	Visual studio 2022	
Dell XPS 15	i7, up to 4.5 GHz	
Operating system	Windows 10, 64-Bit	
Process thread count	16	
RAM	32 GB	
GPU	NVIDIA GeForce RTX 3050, 4 GB	
Models to program GPU	CUDA 12.0, cuDNN 9.0.0	

For the experiments, we utilized the TrashNet dataset with the ratio of 75, 15, 10 for training, testing and validation respectively. The details of used dataset, and division for experimental purpose is presented in Table 3. In first step, we developed three smart bins to collect solid waste in three categories including plastic, glass, and metal. The user is directed carefully from first step to place the waste items in relevant bins, otherwise fourth simple bin (trash bin) can be used to place the rest of waste items. From initial step, the system forces the user to put the waste in relevant bin otherwise an alert is generated to indicate that irrelevant waste item is placed in dedicated smart bin. Figures 3 and 4 presents the implementation of smart bins and micro-controller respectively.

Table 3 Dataset partition details.

Classes	Training (85%)	Testing (15%)	Total	
Glass	426	75	501	
Metal	348	62	410	
Plastic	409	73	482	
Trash	117	20	137	

Figure 3 Block diagram of proposed intelligent waste collection and segregation system.

Figure 4 Micro-controller.

The configuration in Fig. 5 describes the smart dustbin system designed to efficiently collect and manage waste. At its core, the system uses an Arduino Nano microcontroller to control operations, with an HC-SR04 ultrasonic sensor mounted on the garbage bin to measure waste levels. The ultrasonic sensor sends a trigger pulse and receives an echo signal to determine the distance to the waste, enabling real-time garbage bin level sensing. This data is transmitted via UART communication (TX/RX) to an ESP8266 WiFi adaptor operating in STA mode, which connects the system to a Wireless LAN. This connectivity allows the dustbin to send waste level updates to a central monitoring system or a mobile application, facilitating timely waste collection and reducing overflow. The system is powered by a battery, ensuring portability and flexibility in deployment. Overall, this configuration integrates sensor subsystems, embedded systems, and network subsystems to create an intelligent waste management solution that enhances efficiency and sustainability.

Figure 5 Configuration of IoT sensors and the integration between different system components in smart bin.

Once the smart bin(s) is filled according to defined threshold values, waste collector gets the bin information along with location through mobile application, collect the waste and place it at particular dump area. In next phase, waste segregation process is started in refined manners. The implementation of segregation platform is shown in Fig. 6 as follows.

Figure 6 Mechanical model for segregation process of waste items.

Datasets description

This research employs the TrashNet dataset, accessible on GitHub, comprising around 3.5 GB of manually gathered data. The dataset encompasses six classifications including metal, paper, glass, cardboard, and trash, encompassing 99% of recyclable materials. At present, it encompasses 2,527 waste images, each with dimensions of 512 × 384 pixels, adjustable via resizing. These images were captured with items positioned on a white poster board under room illumination or natural light. Figure 7 showcases representative samples from each category within the dataset.

Figure 7 Sample images used in TrashNet dataset.

Data preprocessing: To enhance the quality of data input and improve model performance, we applied the following preprocessing steps: Resizing: All images were resized to 224 × 224 pixels to standardize input dimensions for VGG-19, ResNet, and MobileNet V2.

Normalization: Pixel values were scaled to the range [0,1] to ensure numerical stability and accelerate convergence during training.

Grayscale to RGB conversion: If an image was in grayscale, it was converted to a three-channel RGB format to match the expected input format of deep learning models.

Noise reduction: We applied Gaussian smoothing to reduce unwanted noise without affecting important waste features.

Data augmentation techniques: To address dataset limitations and improve model generalization to real-world conditions, we implemented a data augmentation pipeline that artificially expands the dataset by introducing variations commonly encountered in waste collection environments: Random rotation (±30°): Simulates different orientations of waste objects.

Brightness and contrast adjustments (±20%): Mimics varying lighting conditions.

Horizontal and vertical flipping: Increases diversity by mirroring waste items.

Occlusion simulation (random erasing): Introduces random occlusions to make the model robust against partial visibility.

Background randomization (CutMix and MixUp): Combines different waste items to simulate complex real-world waste piles.

Handling class imbalance: TrashNet, like most real-world waste datasets, exhibits class imbalance, where certain waste categories (e.g., plastic and paper) have significantly more samples than others (e.g., metal and glass). To mitigate this, we adopted the following strategies: Weighted loss function: We used a class-weighted cross-entropy loss, where underrepresented classes were assigned higher weights to ensure balanced learning.

Oversampling & synthetic data generation: We applied Synthetic Minority Over-sampling Technique (SMOTE) to generate synthetic samples for underrepresented waste categories.

Batch balancing: During training, we implemented a stratified sampling approach to ensure each batch contained a proportionate mix of waste types, preventing the model from being biased toward majority classes.

For this research, a portion of the dataset focusing on plastic, metal, glass, and trash was employed. This subset was divided into two parts: one for training, and the other for testing, with proportions of 85% and 15%, respectively. However, to assess the model in real-time, the entire dataset was used for training, and testing was performed on images obtained in real-time. Table 3 presents the distribution of samples across categories in each subset, revealing variations in sample quantities per category.

Performance evaluation metrics

Accuracy

Accuracy (Acc) is a common metric used to evaluate the performance of a classification model. It measures the proportion of correctly classified instances out of the total instances evaluated. Acc can be calculated by counting the number of predictions that match the actual labels and divide by the total number of predictions made. Let TP denote the number of true positive predictions (correctly classified instances), TN denote the number of true negative predictions (correctly rejected instances), FP denote the number of false positive predictions (incorrectly classified instances), and FN denote the number of false negative predictions (incorrectly rejected instances). The accuracy is calculated using Eq. (1) as follows:

(1) ACC=TP+TNTP+TN+FP+FN.

Precision

Precision (P) is a measure used in deep learning to quantify the accuracy of a classification algorithm. It measures the proportion of true positive instances among all instances classified as positive by the model. In other words, precision tells us how many of the items identified as positive are actually relevant. Eq. (2) presents the mathematical form of precision as follows:

(2) P= TPTP+FP.

Average precision

Average precision (AP) is second metric used to evaluate information retrieval systems of iSSWMs, particularly in tasks such as document retrieval or recommendation systems. It calculates the average precision across multiple queries or instances. For each query or instance: Calculate precision at each relevant document position.

Precision at a given position k is the ratio of relevant documents retrieved up to position k to the total number of documents retrieved up to that position.

If there are no relevant documents in the retrieved set, precision at that position is defined to be 0.

The formula for AP is:

(3) AP=∑kN⁡(P(k)∗rel(k))N.

Loss function

A loss function (LF), also known as a cost function or objective function, that quantifies the difference between the predicted values of a model and the actual ground truth values. It serves as a measure of how well the model is performing on a given task. Therefore, LF can be calculated by following the Eq. (4) as follows:

(4) LF=Errorcls+Errorcoord+ErrorIoU.

Results

This section presents the evaluated results of our proposed model, and comparison with existing state-of-the-art studies. We observe that our proposed iSSWMs reveals significant advancements for solid waste classification. Existing studies have employed architecture like ResNet, SVM, VGG-16, ResNet, AlexNet, DenseNet, MobileNet, and RCNN, focusing on different types of solid waste such as glass, metal, paper, plastic, and recyclable waste. While these methods achieved respectable accuracies ranging from 84% to 95%, the iSSWMs, utilizing VGG-19, ResNet, and MobileNet V2 architectures, demonstrated remarkable performance improvements across multiple waste types, including glass, plastic, metal, and general trash. Specifically, the iSSWMs achieved accuracy of 99.7%, 97.2%, and 98.18% for VGG-19, ResNet, and MobileNet V2, respectively, showcasing their superior effectiveness in solid waste classification tasks compared to existing methodologies as shown in Table 4.

Table 4 Accuracy comparison of proposed iSSWMs with existing state-of-the-art methods.

Method	Used architecture	Solid waste types	Results (Acc)	
Adedeji & Wang (2019)	ResNet, SVM	Glass, metal, paper, plastic	ResNet: 87%	
SVM: 84%	
Lilhore et al. (2024)	VGG-16, ResNet, AlexNet	Organic waste	VGG-16: 87%	
ResNet: 88%	
AlexNet: 94%	
Ahmed et al. (2023)	DenseNet, MobileNet, ResNet	Recyclable waste	DenseNet:91%	
MobileNet: 93%	
ResNet: 95%	
Vaidya & Paunwala (2019)	RCNN, F-CNN, SSD	Glass, plastic	RCNN: 90%	
F-CNN: 92%	
SSD: 94.8%	
Shahab & Anjum (2022)	MP-CNN (Multi-Path CNN)	Recyclable solid waste	MP-CNN: 96%	
Krishna & Sharma (2023)	VGG-16, ResNet-16	Metal, paper, plastic	VGG-16: 94%	
ResNet-16: 96.88%	
Sheng et al. (2020)	LoRa	Glass, plastic	LoRa: 97.26%	
iSSWMs
(Ours)	VGG-19, ResNet, MobileNet V2	Glass, plastic, metal, trash	VGG-19: 99.7%	
ResNet: 97.2%	
MobileNet V2: 98.18%	

Further we evaluated average precision against each waste item, and observed notable differences emerge. In previous studies (Adedeji & Wang, 2019; Ahmed et al., 2023; Vaidya & Paunwala, 2019), the AP generally increased with the number of epochs, reaching maximum values ranging from 79% to 92%. However, the proposed iSSWMs model consistently outperformed these existing techniques across all epochs and waste items (glass, plastic, and metal). As the number of epochs increased, the AP of the iSSWMs model exhibited substantial improvements, surpassing 99% with just 80 epochs and reaching an exceptional accuracy of 99.7% with 100 epochs as shown in Figs. 8–10, respectively.

Figure 8 AP against solid waste item glass.

Figure 9 AP against solid waste item plastic.

Figure 10 Average precision against solid waste items (metal).

Similarly, we evaluated recall measures the ability of a model to correctly identify all relevant instances (true positives) out of all actual positives. We also evaluated F1-score which is the harmonic mean of precision and recall. F1-score is useful when you need to strike a balance between precision and recall as shown in Table 5. It is especially valuable in situations where both false positives and false negatives are important, and the class distribution is imbalanced. Recall and F1-score are calculated as in Eqs. 5 and 6 respectively:

Table 5 Recall and F1-score of proposed solution and existing studies.

Method	Used architecture	Recall (%)	F1-score (%)	
Adedeji & Wang (2019)	ResNet, SVM	ResNet: 85%,	ResNet: 86%,	
SVM: 82%	SVM: 83%	
Lilhore et al. (2024)	VGG-16, ResNet, AlexNet	VGG-16: 85%,	VGG-16: 86%,	
ResNet: 86%,	ResNet: 87%	
AlexNet: 92%	AlexNet: 93%	
Ahmed et al. (2023)	DenseNet, MobileNet, ResNet	DenseNet: 89%,	DenseNet: 90%,	
MobileNet: 91%,	MobileNet: 92%,	
	ResNet: 93%	ResNet: 94%	
Vaidya & Paunwala (2019)	RCNN, F-CNN, SSD	RCNN: 88%,	RCNN: 89%,	
F-CNN: 90%,	F-CNN: 91%,	
SSD: 93%	SSD: 94%	
Shahab & Anjum (2022)	MP-CNN	MP-CNN: 94%	MP-CNN: 95%	
Krishna & Sharma (2023)	VGG-16, ResNet-16	VGG-16: 92%,	VGG-16: 93%,	
ResNet-16: 95%	ResNet-16: 96%	
Sheng et al. (2020)	LoRa	LoRa: 96%	LoRa: 97%	
iSSWMs (Our)	VGG-19, ResNet, MobileNet V2	VGG-19: 98%,	VGG-19: 99%,	
ResNet: 95%,	ResNet: 96%,	
		MobileNet V2: 97%	MobileNet V2: 98%	

(5) Recall=Truepostive(TP)Truepostive(TP)+FalseNegtives(FN)

(6) F1=2XprecisonXrecallprecison+recall

Further, we evaluate the accuracy and loss rate of our proposed iSSWMs model while training and testing. Figures 11, and 12 show the fitting process of iSSWMs model, displaying the accuracy and loss curves. Both graphs indicate the absence of overfitting, as the test curves consistently show an uptrend in accuracy or a downtrend in loss, mirroring the behavior of the training curves. This indicates the model’s effective learning in accurately classifying the data, extracting distinctive features across various stages, and utilizing them for class identification during both training and testing phases. Furthermore, it presents robustness against overfitting to the predominant class (middle), notwithstanding the intrinsic class imbalance within the dataset.

Figure 11 Accuracy of proposed iSSWm model while training and test accuracy.

Figure 12 Loss rate of proposed iSSWm model.

Discussion

The article introduces a pioneering methodology, iSSWMs, crafted to optimize the collection and categorization of solid waste for recycling objectives. This method unfolds in two pivotal phases: firstly, leveraging smart bins connected to cloud and mobile applications for the intelligent gathering of household solid waste, encompassing glass, plastic, and metal; and secondly, segregating the gathered waste via sophisticated deep learning technologies integrated with smart mechanical components for recognition. The efficacy of this methodology has been substantiated through validation using the TrashNet dataset and its implementation in real-world scenarios. Notably, in contrast to previous studies, the iSSWMs approach does not require a supplementary workforce, thereby heightening its practicality and cost-effectiveness. To bolster results compared to cutting-edge techniques, the VGG-19 algorithm is enlisted. Renowned for its computational efficiency and accuracy in object detection and classification, this algorithm features 19 layers, comprising 16 convolutional layers and three fully connected layers. VGG-19 adeptly extracts pertinent features pivotal for precise classification. These convolutional layers compute salient features weighted by a deep learning algorithm, significantly contributing to the overall performance metrics of the proposed system. Training the dataset with the VGG-19 deep learning algorithm for 100 epochs, each representing a comprehensive training cycle, attains a remarkable accuracy of 99.7%. Technically our proposed iSSWMs model surpasses existing techniques due to the superior architectural advantages of VGG-19, ResNet, and MobileNet V2, which enhance feature extraction, classification accuracy, and computational efficiency. VGG-19 outperforms earlier deep learning models like VGG-16 and AlexNet by utilizing deeper convolutional layers (19 layers), which enable more refined hierarchical feature extraction. This deeper architecture allows iSSWMs to achieve 99.7% accuracy, significantly higher than the 94% reported in previous VGG-16-based studies. Similarly, MobileNet V2, designed for efficient depthwise separable convolutions, enhances computational efficiency while maintaining high classification accuracy. Unlike heavy architectures like DenseNet and RCNN, which demand significant processing power, MobileNet V2 achieves a competitive 98.18% accuracy with significantly reduced computational overhead, making it ideal for real-time waste classification in IoT-driven applications. On other hand, ResNet leverages residual learning and skip connections, which effectively mitigate the vanishing gradient problem in deep networks. Compared to traditional CNNs, ResNet achieves better generalization and higher accuracy (97.2% in our study vs. 96.88% in previous work), demonstrating its robustness in complex waste classification tasks.

Conclusion

The advent of advanced technologies such as AI, IoT, and ML has opened new avenues for addressing the pressing issue of solid waste management. Our proposed framework, an intelligent and smart solid waste management system, represents a significant step towards revolutionizing waste management practices by integrating these cutting-edge technologies. Through the deployment of smart bins equipped with IoT sensors and cloud-based analytics, coupled with deep learning techniques for waste segregation, iSSWMs offers a comprehensive solution for efficient waste collection and recycling. The results demonstrate the efficacy of the iSSWMs framework, with remarkable accuracy rates achieved in waste classification tasks. Leveraging architectures like VGG-19, iSSWMs outperforms existing methodologies across multiple waste types, including glass, plastic, metal, and general trash with maximum accuracy level 99.7%. Moreover, the practicality and cost-effectiveness of iSSWMs are underscored by its ability to operate without supplementary workforce, making it a scalable solution for both urban and rural areas. By streamlining the waste collection and segregation process, iSSWMs not only contributes to environmental sustainability but also addresses public health concerns by reducing the spread of diseases associated with improper waste disposal. In essence, iSSWMs represents a promising framework for modernizing waste management systems, offering a scalable, efficient, and technology-driven approach to tackling the global challenge of solid waste management. As we continue to refine and implement such innovative solutions, we move closer to achieving a cleaner and healthier environment for present and future generations.

Supplemental Information

Supplemental Information 1 Thinkspeak data for Smart dustbin I.

Supplemental Information 2 Thinkspeak data for Smart dustbin II.

Additional Information and Declarations

Competing Interests

The authors declare that they have no competing interests.

Author Contributions

Abdullah Alourani conceived and designed the experiments, performed the experiments, analyzed the data, performed the computation work, prepared figures and/or tables, authored or reviewed drafts of the article, and approved the final draft.

M. Usman Ashraf conceived and designed the experiments, performed the experiments, analyzed the data, performed the computation work, prepared figures and/or tables, authored or reviewed drafts of the article, and approved the final draft.

Mohammed Aloraini conceived and designed the experiments, performed the experiments, performed the computation work, prepared figures and/or tables, authored or reviewed drafts of the article, and approved the final draft.

Data Availability

The following information was supplied regarding data availability:

The code is available at GitHub:

https://github.com/usmanashraf88/swms/blob/main/main_bin_seg.

https://github.com/usmanashraf88/swms/blob/main/thinkspeak.ino.

https://github.com/usmanashraf88/swms/blob/main/ml_model.sav.

The dataset is available at GitHub:

https://github.com/garythung/trashnet.

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
