# Peer review of "Smart waste management and classification system using advanced IoT and AI technologies"

_PeerJ Computer Science, doi:10.7717/peerj-cs.2777_

## Round 0.1 · original submission · Major Revisions

Dear Authors,
Your paper has been revised. Based on the reviewers' evaluation, minor revisions are needed before it is considered for publication in PEERJ Computer Science. More precisely, the following points must be faced in the revised version of your paper:

1) The literature review section must be improved, including the recent advancements in IoT-enabled smart waste management systems.
2) A precise comparative analysis of the proposed approach against all relevant state-of-the-art systems, especially those employing similar IoT and AI methodologies must be added.
3) Additional datasets and real-world deployment scenarios to validate the system's robustness must be considered.

·

Basic reporting

The manuscript proposes an Intelligent and Smart Solid Waste Management System (iSSWMs) integrating IoT and AI for efficient waste collection and segregation with the VGG-19 model. The concept is okay, but several similar studies are available. The statement of its novelty is questionable, and it lacks a comprehensive review of existing works and a critical discussion of their limitations to justify the novelty of the proposed work. These are some specific limitations that need to be addressed:
1. The literature review section does not adequately cover recent advancements in IoT-enabled smart waste management systems. The authors should include more recent, relevant works and highlight how their contributions surpass these.
2. The authors claim that their proposed iSSWMs framework is groundbreaking without providing a clear comparative analysis against all relevant state-of-the-art systems, especially those employing similar IoT and AI methodologies.
3. Provide a more detailed analysis and justification for the chosen methodologies compared to existing systems.
4. The study uses the TrashNet dataset, which may not fully represent real-world scenarios.
5. Include additional datasets and real-world deployment scenarios to validate the system’s robustness.
6. Although the TrashNet dataset is used, the authors fail to provide sufficient details about preprocessing steps, augmentation techniques, or handling of class imbalances, which are critical for reproducibility and robustness.
7. The study predominantly highlights accuracy, but other critical metrics such as F1-score, recall, and precision for classifying underrepresented categories are not discussed in depth.
8. Extend the metrics evaluated to include recall, F1-score, and confusion matrices.
9. There is no analysis of the system's scalability to larger datasets or its robustness under varying conditions such as lighting, object orientation, or overlapping items.
10. The methodology lacks clarity in certain technical aspects, such as the configuration of IoT sensors and the integration between different system components. Specifics about hardware and software integration are minimal.
11. Clarify the system's integration process and scalability potential in diverse environments.

Experimental design

No comments

Validity of the findings

No comments

Additional comments

No comments

·

Basic reporting

The submission adhered to PeerJ Computer Science policies
Yes, the article uses mostly clear and professional language.

Literature references, field background, and context:
The literature review cites relevant works related to IoT and AI in waste classification. However, the review could incorporate more recent and high-profile journal articles to ensure comprehensive coverage of advancements in AI and IoT in waste management, and missing context on prior studies about transfer learning and IoT scalability in this domain could also be added.

Professional structure, figures, and tables:
Figures (e.g., IoT setup, sample images) were adequately labeled. More detailed captions and descriptions should be done to enhance the interpretation of tables and figures. Also use a higher resolution for some of the figures (e.g., IoT setup photo) to improve the readability.

Self-contained with relevant results to hypotheses:
Some aspects of the results, like error analysis or statistical validation, should be thoroughly addressed.

Not all terms used in the formal results were defined. All terms used in the results should be clearly defined and detailed proofs shown.

Experimental design

The Study falls within the aims and Scope of the Journal. The research presents original work, leveraging IoT and AI for smart waste management.

The research question is meaningful and aligns with a pressing global issue. It addresses how IoT and AI can improve waste sorting efficiency.
After comparing the current methodology to prior work, the study should explicitly state how it fills the identified knowledge gap.

Rigorous investigation:
The use of AI-based models and IoT devices suggests a high technical standard, the use of image-based waste classification is innovative and appropriate for the problem.

Ethical considerations related to data collection (e.g., images used in training) should be mentioned.

Methods described with replicability:
The methodology includes key steps, but insufficient detail is provided for replication:

There is no mention of steps taken to control bias or ensure reproducibility.

No specific details on preprocessing steps, hyperparameter tuning, or model training duration.
Hardware limitations and software configurations are underexplained.

Validity of the findings

The study should give a detailed experimental procedure on how it used transfer learning, a well-accepted and replicable method.

The results are summarized, but statistical soundness is not explicitly verified. Details on statistical measures like confidence intervals, p-values, or error bars should be given.

The conclusions align with the hypothesis and findings and the the results are directly tied to the problem of waste sorting and IoT deployment.

Future work could be stated to handle the limitations of the study. The conclusions would benefit from deeper analysis of limitations and potential future work, particularly on the system's scalability and accuracy in diverse conditions.

Additional comments

Suggestions for Improvement
1. Enhance Methodological Rigor:
Include alongside your dataset size, diversity and source.
Provide more details about preprocessing, hyperparameters, and training processes.
Include the error analysis, confidence intervals, and detailed metrics like F1 scores in the study.

Improve Clarity and Flow:
Revise sentences for grammatical correctness and coherence using grammatical tools.
Ensure consistent terminology and a professional tone.

Consider including a brief note in the figure caption about how the dataset is preprocessed for the classification model.
Ensure image quality is consistent across all examples to maintain visual appeal


Incorporate more references:
Add recent, high-impact studies from journals like IEEE Transactions on IoT, and Nature Sustainability, to enrich the study.

Occasional grammatical errors and awkward sentence structures are present. For example:
Some sections lack transitional sentences, which affects flow and coherence.
Terminology like "recyclable materials" could benefit from consistent use to avoid ambiguity.

Ensure all terms are defined when first introduced, such as "TrashNet."
Figures: Improve descriptions and align them with the article's main text.

Conduct a final proofreading pass to catch minor grammatical issues, such as missing articles or verb agreement errors. for example: Replace vague statements with precise ones:

Original study statement: “The system is better than other methods.”
Suggested statement: “The system outperforms existing methods in terms of accuracy and scalability.”

---

## Round 0.2 · accepted · Accept

Dear Authors,

Your paper has been revised. It has been accepted for publication in PEERJ Computer Science. Thank you for your fine contribution.

·

Basic reporting

This is a re-review of the article that was submitted.
the article uses mostly clear and professional language.

The Author has included three most recent and published well-known journal articles [47], [48], [49]. and also included a comparative analysis table in the literature review section for a quick overview of existing state-of-the-art methods. This is okay.

The figures and tables are well displayed.


All terms used are well-defined

No further comments..

Experimental design

Corrections have been done.


No comment.

Validity of the findings

the corrections have been done.

No comment.

Additional comments

Your corrections are accepted.